# Defining the biological basis of radiomic phenotypes in lung cancer

Patrick Grossmann[1,2], Olya Stringfield[3], Nehme El-Hachem[4], Marilyn M Bui[5], Emmanuel Rios Velazquez[1], Chintan Parmar[1,6], Ralph TH Leijenaar[6], Benjamin Haibe-Kains[7,8], Philippe Lambin[6], Robert J Gillies[3], Hugo JWL Aerts[1,2,9]*

[1]Department of Radiation Oncology, Dana-Farber Cancer Institute, Brigham and Women's Hospital, Harvard Medical School, Boston, United States; [2]Department of Biostatistics and Computational Biology, Dana-Farber Cancer Institute, Boston, United States; [3]Department of Cancer Imaging and Metabolism, H. Lee Moffitt Cancer Center and Research Institute, Tampa, United States; [4]Integrative systems biology, Institut de recherches cliniques de Montreal, Montreal, Canada.; [5]Department of Anatomic Pathology, H. Lee Moffitt Cancer Center and Research Institute, Tampa, United States; [6]Department of Radiation Oncology, Research Institute GROW, Maastricht University, Maastricht, Netherlands; [7]Princess Margaret Cancer Centre, University Health Network, University of Toronto, Toronto, Canada; [8]Medical Biophysics Department, University of Toronto, Toronto, Canada; [9]Department of Radiology, Brigham and Women's Hospital, Harvard Medical School, Boston, United States

*For correspondence: hugo_aerts@dfci.harvard.edu

**Abstract** Medical imaging can visualize characteristics of human cancer noninvasively. Radiomics is an emerging field that translates these medical images into quantitative data to enable phenotypic profiling of tumors. While radiomics has been associated with several clinical endpoints, the complex relationships of radiomics, clinical factors, and tumor biology are largely unknown. To this end, we analyzed two independent cohorts of respectively 262 North American and 89 European patients with lung cancer, and consistently identified previously undescribed associations between radiomic imaging features, molecular pathways, and clinical factors. In particular, we found a relationship between imaging features, immune response, inflammation, and survival, which was further validated by immunohistochemical staining. Moreover, a number of imaging features showed predictive value for specific pathways; for example, intra-tumor heterogeneity features predicted activity of RNA polymerase transcription (AUC = 0.62, p=0.03) and intensity dispersion was predictive of the autodegration pathway of a ubiquitin ligase (AUC = 0.69, $p<10^{-4}$). Finally, we observed that prognostic biomarkers performed highest when combining radiomic, genetic, and clinical information (CI = 0.73, $p<10^{-9}$) indicating complementary value of these data. In conclusion, we demonstrate that radiomic approaches permit noninvasive assessment of both molecular and clinical characteristics of tumors, and therefore have the potential to advance clinical decision-making by systematically analyzing standard-of-care medical images.

## Introduction

'Precision medicine' promotes the molecular characterization of a patient's tumor with genomic approaches, which requires tissue extraction usually obtained via biopsy. A number of examples demonstrate successful translation of genomic information obtained from biopsies into clinical applications (*Doroshow and Kummar, 2014*), but these approaches also have inherent limitations, such

**eLife digest** Medical imaging covers a wide range of techniques that are used to look inside the body, including X-rays, MRI scans and ultrasound. A process called radiomics uses computer algorithms to process the data collected by these techniques to identify and precisely measure a large number of features that would not otherwise be quantifiable by human experts. By doing so, radiomics can automatically measure the radiographic characteristics of a tumor. For example, radiomics can establish the size, shape and texture of a tumor to help to diagnose cancer and guide its treatment.

Research has suggested that radiomics can predict certain clinical characteristics of cancer, such as how far through the body the cancer has spread, how likely it is to respond to treatment, and how likely a patient is to survive. However, these radiomic characteristics have not yet been precisely linked to the biological processes that drive how cancer develops and spreads.

Cancers develop as a result of genetic changes that activate "molecular pathways" in the cells and trigger processes such as cell division and inflammation. To work out exactly which changes are behind a particular tumor, a sample of the tumor from biopsy or surgery is analyzed using genomics techniques. Linking radiomics features to the molecular processes active in a tumor can generate further information that can complement the molecular data. Images are routinely collected on all cancer patients yet molecular data is not. Hence, in some cases, the images can be used to infer the molecular underpinnings of cancer in individual patients.

Grossmann et al. have now analyzed radiomic, genomic and clinical data collected from approximately 350 patients with lung cancer. The analysis revealed links between biological processes normally detected by genomics – in particular, inflammatory responses – and radiomics features. Furthermore, these features could also be associated with clinical characteristics, such as tumor type and patient survival rates. These results were further validated by using a technique called immunohistochemical staining on tumor tissue obtained by surgery.

Further investigation revealed that certain radiomics features can predict the state of molecular pathways that are key to cancer development (such as the inflammatory response). Furthermore, Grossmann et al. found that combining data from radiomics, genomics and clinical parameters predicts how the cancer will progress better than any of these parameters can predict on their own. These results demonstrate the complementary value of radiomic data to genomic and clinical data.

There are many different algorithms that can be used to process images for radiomics. Before radiomics can be used clinically to assess the biological processes underlying the tumors of patients, a specific algorithm needs to be decided upon and then tested in prospective clinical trials.

as their invasive nature or sampling artifacts caused by intra-tumor heterogeneity (*Sottoriva et al., 2013*; *Fisher et al., 2013*; *Gerlinger et al., 2012*). These limitations can be addressed by medical imaging that has served as crucial diagnostic tool and treatment guidance in clinical oncology. In contrast to biopsies, medical imaging is usually noninvasive, can be applied longitudinally, and provides information about the entire visible tumor volume. In this way, medical imaging has the potential to characterize phenotypic information of tumors and thus complement molecular interrogation (*Choi et al., 2016*). As imaging is already used routinely throughout the course of treatment this facilitates ready access to this type of data. Therefore, imaging has the potential to serve as valuable diagnostic tool in clinical decision making by complementing biological interrogation or serving as a surrogate in settings where biospecimen-derived diagnostics is not feasible, such as in longitudinal monitoring.

Radiomics is an emerging field that translates these medical images into mineable data by extracting a large number of quantitative imaging features that objectively define tumor intensity, shape, size, and texture (*Gillies et al., 2016*; *Aerts, 2016*; *Lambin et al., 2012*; *Kumar et al., 2012*) in a robust and reproducible way (*Zhao et al., 2016*; *Fried et al., 2014*; *Balagurunathan et al., 2014*; *Leijenaar et al., 2013*). As this approach is applied to existing standard of care images, radiomics can be cost-effectively integrated with genomics or serve as surrogate in cases where biopsies are not feasible (*O'Connor et al., 2015*). Hence, such strategies can be of value for the development

of clinical biomarkers for diagnosis, prognosis, and prediction of response to specific treatments (*Choi et al., 2016*; *Huang et al., 2016a, 2016b*; *Aerts et al., 2016*; *Nicolasjilwan et al., 2015*; *Parmar et al., 2015a, Parmar et al., 2015b*; *Aerts et al., 2014*; *Chong et al., 2014*; *Coroller et al., 2015*; *Gevaert et al., 2012*; *Ganeshan et al., 2012*; *Win et al., 2013*; *Mattonen et al., 2016*; *Grossmann et al., 2017*). Due to the enormous potential for precision medicine, an increasing number of studies have investigated associations between imaging and tumor biology in different cancer types (*Aerts et al., 2014*; *Gevaert et al., 2012*; *Diehn et al., 2008*; *Grossmann et al., 2016*; *Gutman et al., 2015*; *Segal et al., 2007*; *Li et al., 2016*; *Yoon et al., 2015*). However, these studies focused on specific genetic associations, or tended to be underpowered due to a limited number of available samples and lacked validation via independent datasets.

Here, we present a broad radiomic-genomic analysis in independent and large cohorts of patients with lung cancer. We rigorously investigated the mechanistic connections between imaging phenotypes and underlying molecular pathways. Furthermore, we validated key associations via immunohistochemical staining and related these associations to clinical factors. In addition, we developed and validated radiomic predictors of pathway activation status, and investigated the prognostic value of combining radiomic biomarkers with genetic and clinical data. In this study, we aimed at uncovering whether radiomic approaches have the potential to predict both molecular and clinical characteristics of tumors noninvasively and therefore have the potential to augment clinical decision-making using data extracted from standard of care medical images.

## Results

To uncover the mechanistic connections between radiomic phenotypes, molecular pathways, and clinical information, we performed an integrated radiomic-genomic analysis of a lung cancer discovery cohort (Dataset1, *n* = 262), and validated our results on an independent validation cohort

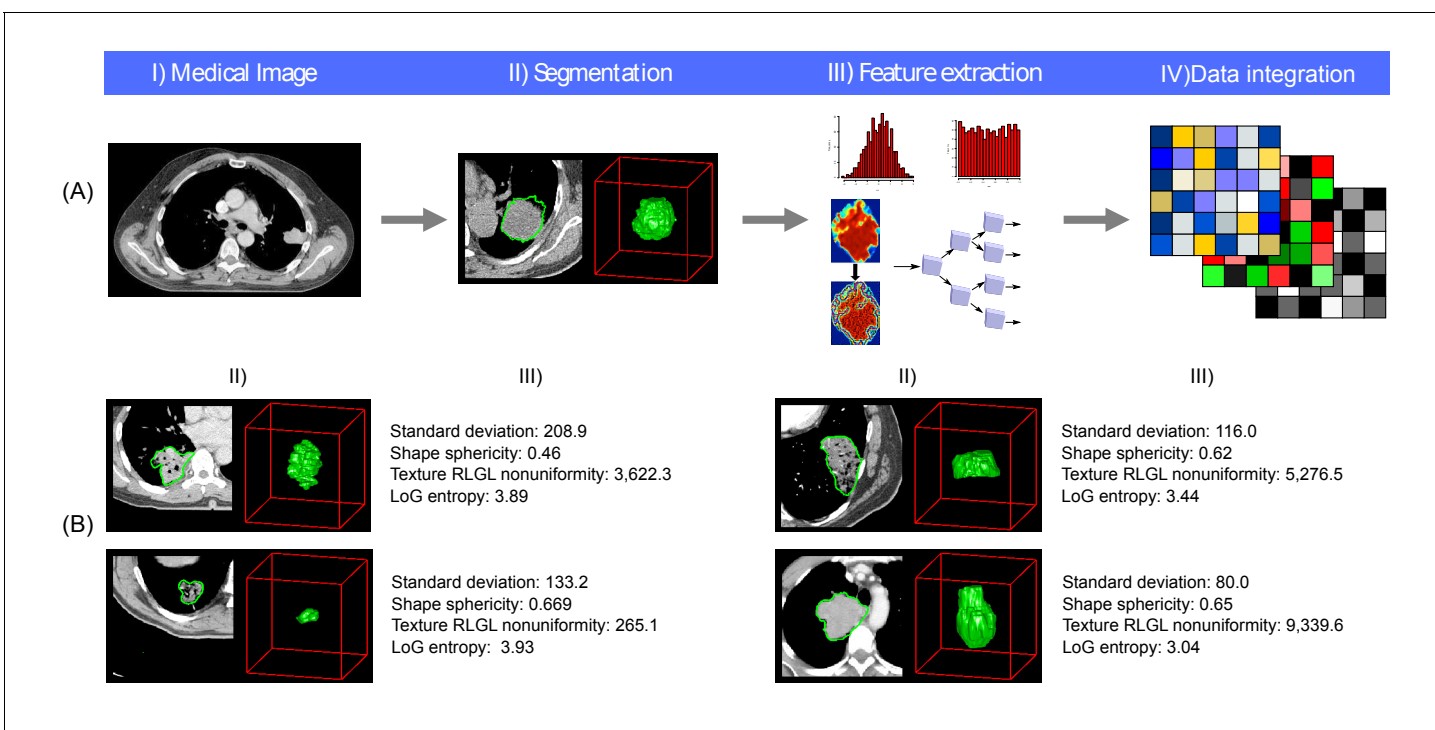

**Figure 1.** Radiomics approach. (**A**) Workflow of extracting radiomic features: (I) A lung tumor is scanned in multiple slices. (II) Next, the tumor is delineated in every slice and validated by an experienced physician. This allows creation of a 3D representation of the tumor outlining phenotypic differences of tumors. (III) Radiomic features are extracted from this 3D mask, and (IV) integrated with genomic and clinical data. (**B**) Representative examples of lung cancer tumors. Visual and nonvisual differences in tumor shape and texture between patients can be objectively defined by radiomics features, such as entropy of voxel intensity values ('How heterogeneous is the tumor?') or sphericity of the tumor ('How round is the tumor?').

(Dataset2, $n$ = 89). We defined and extracted 636 radiomic features from CT scans (*Figure 1A*) quantifying tumor intensity, shape, and texture (*Figure 1B*), detailed in *Supplementary file 1*. Our approach to integrate radiomic, genomic, and clinical data is outlined in *Figure 2* and clinical cohort characteristics are given in *Table 1*. Dataset1 can be downloaded as *Figure 2—source data 1* and Dataset2 can be downloaded as *Figure 2—source data 2*.

## Association modules of radiomic features and molecular pathways

To investigate the main associations of radiomics and underlying molecular pathways, we developed association modules describing radiomic-pathway coherency. Bi-clustering allowed simultaneous grouping of coherently expressed features and pathways into a single module, thereby reducing dimensionality. Using this approach, we identified thirteen radiomic-pathway modules in Dataset1 that were independently validated in Dataset2 (FDR < 0.05). *Figure 3A* and *Table 2* summarize these modules, while a detailed version of every module is given in *Figure 3—source data 1*.

In general, we found that distinct radiomic features were associated with distinct biological processes. For example, texture entropy and cluster features, as well as voxel intensity variance features were associated with the immune system, the p53 pathway, and other pathways involved in cell cycle regulation in modules M2, M9, and M12 (*Table 2* and *Figure 3A*). In another module (M8), we found those features to also be associated with transforming growth factor beta (TGF-$\beta$) receptor signaling.

Further examples for radiomic-pathway links included two modules (M13 and M7) that were highly enriched for pathways involved in mitochondrial pathways, transcription, translation, and RNA regulatory mechanisms; with only one exception, all features in the larger module (M13) were voxel intensity entropy features. In addition to this feature type, the smaller module (M7) contained mainly textural variance and information correlation features.

## Clinical information contained in modules

For every module, we assessed prognostic association to overall survival (OS) and associations to stage and histology based on the radiomic features of a module (*Figure 3B* and *Table 3*). Three modules (M2, M9, and M12) were significantly prognostic for OS ($p<0.02$), ten modules (M2, M4-8, and M10-13) were significantly associated with stage ($p<0.01$), and five modules (M5, M6, and M10-12) were significantly associated with histology ($p<0.05$). The exact p-values of all modules are given in *Supplementary file 2*.

We examined and summarized the relationships of clinical status, module size, and overlap of modules in a network (*Figure 3B* and *Table 3*). We found that smaller modules tended not to be associated with the tested clinical factors. The total number of shared features or pathways was generally low (mean Jaccard index 0.22, range [0.01, 0.59]). Interestingly, certain modules with higher overlap still showed different clinical associations.

## Radiomic predictors of pathway status

To test whether radiomic features can predict if a pathway is activated or deleted in individual patients, we fitted univariate models of radiomic features on Dataset1 and selected for every module the strongest predictor in Dataset1 according to the area under the curve (*Fawcett, 2006*) (AUC) for validation in Dataset2. As shown in *Table 3* and *Table 3—source data 1*, the overall biological and radiomic themes in a module were well represented by these individual predictors. For example, a Laplace of Gaussian intensity standard deviation feature was predictive of the autodegration pathway of the E3 ubiquitin ligase COP1 (AUC = 0.69, $p<10^{-4}$) in module M2, which was also associated with p53. Importantly, COP1 mediates p53 and may interact with autophagy (*Rabbani et al., 2014*; *Kobayashi et al., 2013*), which are known drivers of tumorigenesis. Indeed, this module M2 was associated with OS. We found further examples of this radiomic-genomic-clinical link to be important: For example, a texture feature (information correlation) predicted trafficking of GLUR2 containing AMPA receptors (AUC = 0.69, $p<10^{-4}$) in module 5, which was associated with lipoprotein metabolism and stage. Further, two shape features (sphericity and compactness) predicted TRAF6 mediated NFkB activation (AUC = 0.66, $p=0.003$) in module 10, which was also associated with axon guidance and histology.

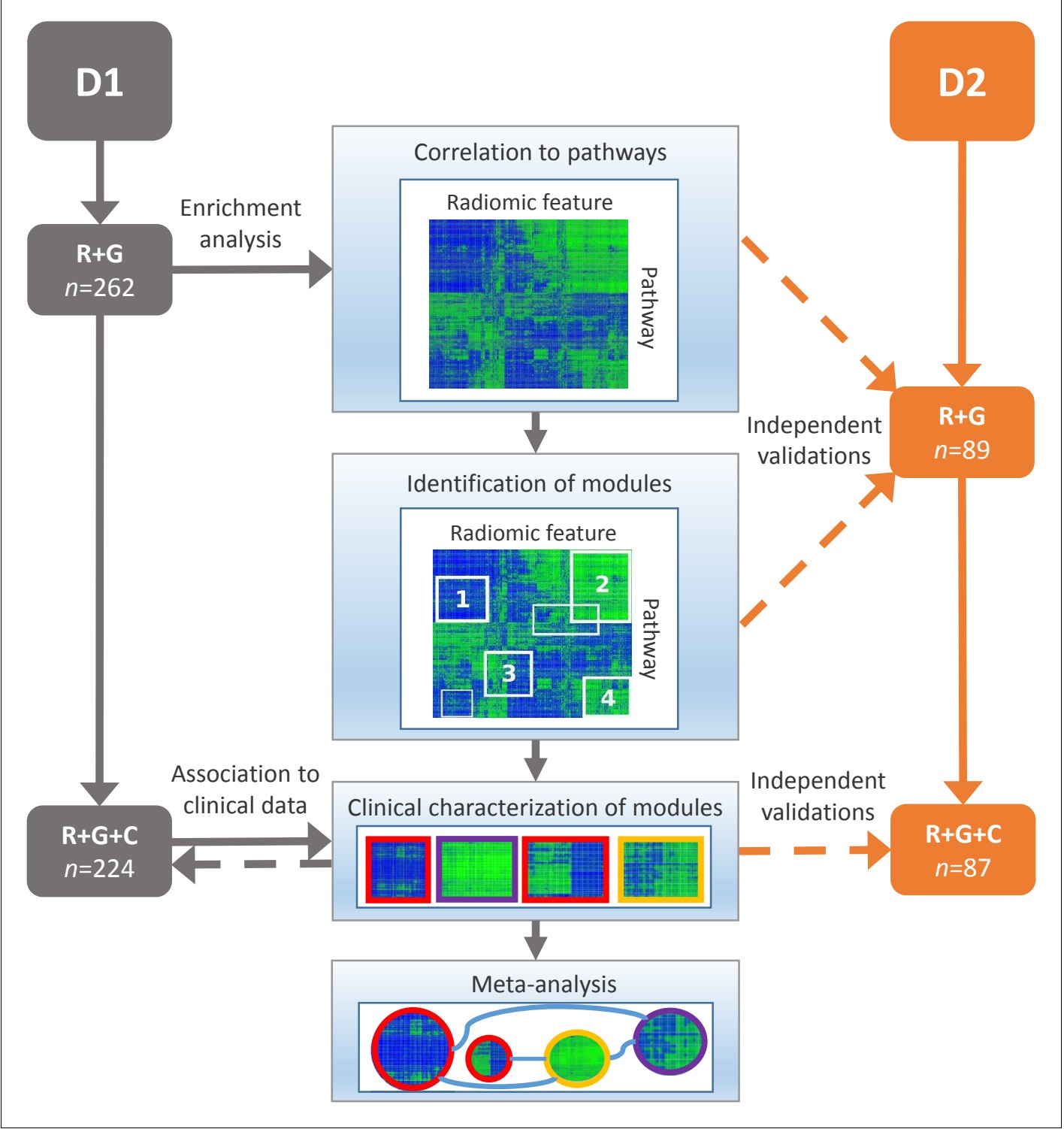

**Figure 2.** Schema of our strategy to define robust radiomic-pathway-clinical relationships. Two independent lung cancer cohorts (D1 and D2) with radiomic (R), genomic (G), and clinical (C) data were analyzed. D1 (n = 262) was used as a discovery cohort and D2 (n = 89) was used to validate our findings. A gene set enrichment analysis (GSEA) approach assessed scores for radiomic-pathway associations. These scores were biclustered to modules that contain features and pathways with coherent expression patterns. These modules may overlap and vary in size. Clinical association to overall survival (red), pathologic histology (purple), and TNM stage (yellow) was statistically tested in both datasets, and results were combined in a meta-analysis to investigate relationships of modules.

*Figure 2 continued on next page*

*Figure 2 continued*

The following source data is available for figure 2:

**Source data 1.** Dataset1.
**Source data 2.** Dataset2.

Furthermore, we assessed these representative features in terms of their predictive value for driver mutations in the discovery cohort; based on a subset of 60 patients whose tumors were profiled with Sanger sequencing, we estimate that the prevalence of mutated EGFR, KRAS, and TP53 are 15%, 35%, and 20%, respectively. In particular, we found strong performance for mutations in EGFR and KRAS by several features, but only one considerable performance for TP53 (*Figure 3—figure supplement 1*). Interestingly, predictive value for EGFR and KRAS were selective in that features had relatively high performance for one gene but not both. Predictive power for smoking history was low to moderate (*Figure 3—figure supplement 2*).

**Table 1.** Proportions of clinical characteristics in Dataset1 and Dataset2, *Figure 2*.
Histology and TNM stage were based on pathology were available.

|  | Dataset1 | Dataset2 |
| --- | --- | --- |
| Gender |  |  |
| Male | 100 (45%) | 59 (68%) |
| Female | 124 (55%) | 28 (32%) |
| Histology |  |  |
| Adenocarcinoma | 129 (58%) | 42 (48%) |
| Squamous | 61 (27%) | 33 (38%) |
| Other | 34 (15%) | 12 (14%) |
| Stage |  |  |
| I | 123 (55%) | 39 (45%) |
| II | 35 (15%) | 26 (30%) |
| III | 46 (21%) | 12 (14%) |
| Other | 20 (9%) | 10 (11%) |
| Smoking Status |  |  |
| Current | 66 (29%) | NA |
| Former | 141 (63%) | NA |
| None | 17 (8%) | NA |
| Tumor site |  |  |
| Primary | 224 (100%) | 87 (100%) |
| Endpoints |  |  |
| Overall survivals | 134 (60%) | 41 (47%) |
| Overall deaths | 90 (40%) | 46 (53%) |
| Follow up (median months) | 32 | 31 |

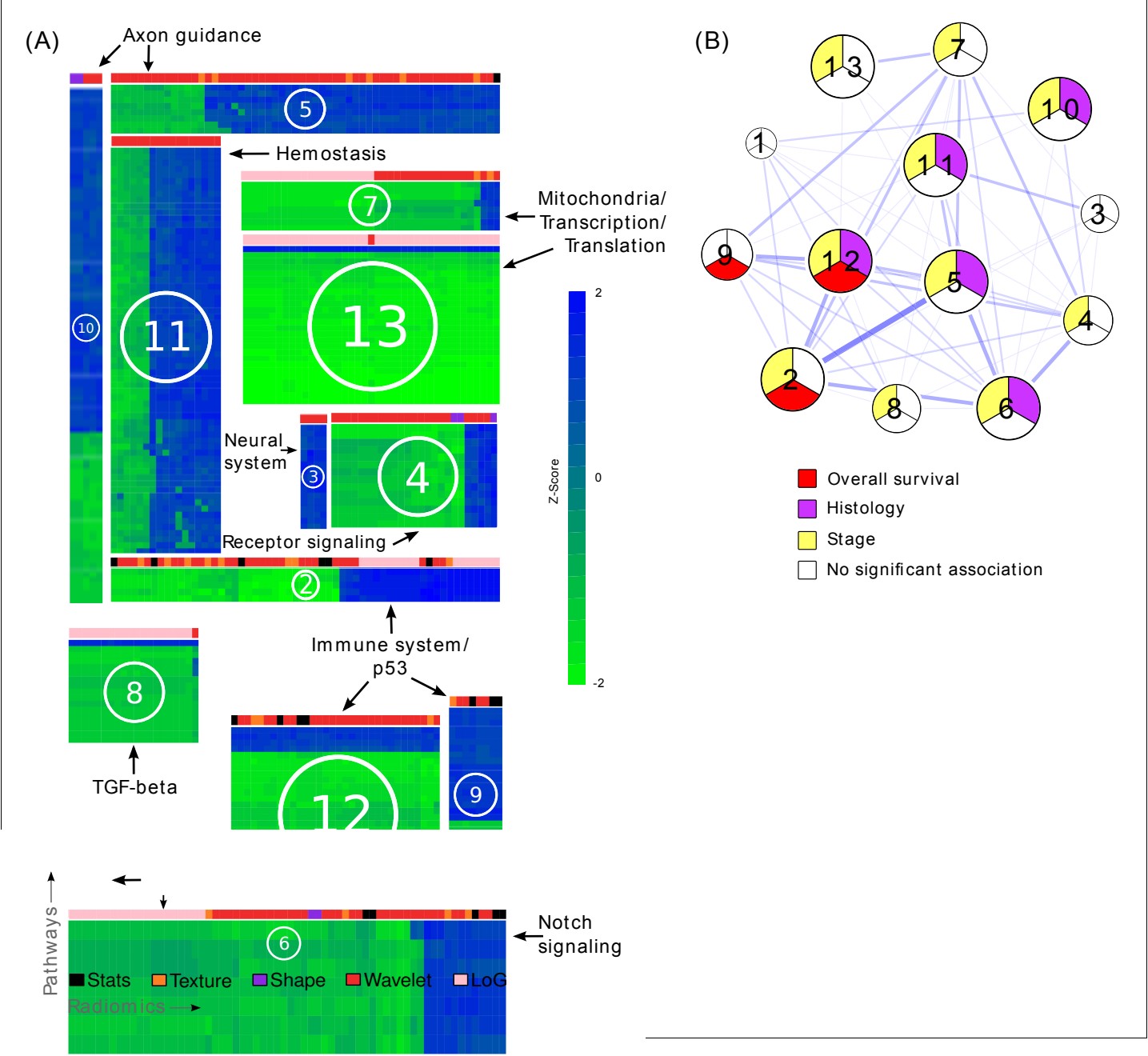

**Figure 3.** Radiomic-pathway-clinical modules. (**A**) Clustering of significantly validated radiomic-pathway association modules (FDR < 0.05). Normalized enrichment scores (NESs) have been biclustered to coherently expressed modules. Every heatmap in this figure corresponds to a module (M1 - M13) with radiomic features in columns and pathways in rows. Heatmap sizes are proportional to module sizes. Elements are NESs given in Z-scores across features, and are displayed in blue when positive and green when negative. Horizontal color bars above every module indicate radiomic feature groups (black = first order statistics, orange = texture, purple = shape, red = wavelet, and pink = Laplace of Gaussian). Representative molecular pathways are displayed. (**B**) Clinical module network. We investigated if modules were associated with overall survival (red), stage (yellow), histology (purple), or no clinical factor (white). Relationships of modules based on their number of shared radiomic features (thickness of blue lines) are displayed by a network. While we found that most modules yield clinical information, overlaps of modules did not indicate relationships to similar clinical factors.

The following source data and figure supplements are available for figure 3:

**Source data 1.** Enlarged heatmaps of every module depicting normalized enrichment scores (NESs) of every pair of radiomic feature and molecular pathway clustered in a module.

*Figure 3 continued on next page*

**Figure supplement 1.** Predictive capabilities of representative radiomic features from every module for genetic mutations in KRAS, EGFR, and TP53 in a subset of the discovery cohort.

**Figure supplement 2.** Association of representative features with smoking history in a subset of the discovery cohort.

## Immunohistochemical investigation

To further investigate putative connections between radiomics, immune response pathways, and OS we performed immunohistochemical staining of 22 tumors for CD3, a T-cell co-receptor. These tumors were predicted to show relatively high or low immune response by a radiomics feature selected from the three modules (M2, M9, and M12) that were associated with OS. As represented in *Figure 4*, we found agreement between radiomics and pathology; cases that were pathologically scored to have high CD3 enrichment also expressed significantly higher radiomic values (one-sided Wilcoxon rank sum test, p=0.008). Furthermore, we tested the extent to which radiomic predictors of inflammation can be reproduced immunohistochemically. We built on our previous results suggesting that the radiomic shape feature sphericity predicts NFkB activation (module 10) and analyzed 24 stained tumors that were predicted to have relatively high or low NFkB activity for RelA, the p65 subunit of NFkB (*Figure 4—figure supplement 1*). Pathological assessment of enrichment for RelA revealed that those cases that indicated high RelA enrichment on average also had higher radiomic feature scores (one-sided Wilcoxon rank sum test, p=0.06).

**Table 2.** Summary of common themes in all of the identified radiomic-pathway association modules. Columns 1–3 display the module name, the number of radiomic features (nr), and pathways (np), respectively. Columns 4–5 hold the radiomic and pathway themes present in each module.

| Module | nr | np | Radiomic | Pathway |
|---|---|---|---|---|
| M1 | 6 | 7 | Wavelet texture gray-level runs | Lipid and lipoprotein metabolism, Notch signaling, circadian clock |
| M2 | 58 | 5 | Wavelet intensity entropy; Laplace of Gaussian intensity standard deviation | Immune system, p53 |
| M3 | 4 | 17 | Wavelet minimum intensity | Neural system, axon guidance |
| M4 | 25 | 14 | Intensity variance and mean; wavelet minimum intensity min | Biological oxidations, signaling by insulin receptor, signaling by GPCR, neuronal system |
| M5 | 58 | 8 | Wavelet texture gray-level runs; wavelet intensity range and median; (wavelet) texture information correlation and cluster tendency | Axon guidance and synaptic transmission, lipoprotein metabolism, cell type determination |
| M6 | 64 | 7 | Laplace of Gaussian standard deviation; wavelet texture gray-level runs; wavelet texture cluster tendency | Circadian clock, signaling by Notch |
| M7 | 39 | 8 | Laplace of Gaussian intensity entropy; wavelet intensity variance; Laplace of Gaussian texture information correlation | Mitochondria, Pol III transcription |
| M8 | 20 | 17 | Laplace of Gaussian standard deviation | TCA cycle and electron transport, TGF-beta receptor signaling, response to stress, transcription regulation, protein synthesis, |
| M9 | 8 | 30 | Intensity variance; wavelet intensity variance | Immune system, p53, cell cycle regulation checkpoints, cell-cell interaction, circadian clock |
| M10 | 5 | 83 | Shape surface (SH); wavelet texture gray-level runs | Axon guidance, neuronal system, (innate) immune system, hemostasis, FGFR signaling, TGF-beta receptor signaling, Notch signaling, circadian clock |
| M11 | 17 | 66 | Wavelet intensity range; wavelet texture information correlation | Hemostasis, neural system |
| M12 | 32 | 27 | Wavelet texture entropy; intensity variance; wavelet texture cluster tendency | P53, immune system |
| M13 | 39 | 26 | Intensity entropy | Gene expression regulation, Pol II/III transcription |

**Table 3.** Pathway prediction and clinical association. For every module, the independent validation performance of the strongest radiomic based pathway predictors is indicated per module by the area under the curve (AUC) of the receiver operator characteristic. In addition, we highlight whether a module was significantly associated with overall survival (OS), TNM stage (ST), or pathologic histology (HI) (p<0.05).

| Module | Strongest radiomic based pathway prediction | AUC | OS | ST | HI |
|---|---|---|---|---|---|
| M1 | Wavelet (HHH) texture (GLCM) correlation → Cholesterol biosynthesis | 0.64, p=0.014 | | | |
| M2 | Laplace of Gaussian intensity standard deviation → Autodegration of the E3 Ubiquitin ligase COP1 | 0.69, p=8e-4 | x | x | |
| M3 | Wavelet minimum intensity → Trafficking of GLUR2 containing AMPA receptors | 0.67, p=0.003 | | | |
| M4 | Wavelet intensity minimum → Glutathione conjugation | 0.68, p=9e-4 | | x | |
| M5 | Texture information correlation → Trafficking of GLUR2 containing AMPA receptors | 0.69, p=7e-4 | | x | x |
| M6 | Wavelet texture cluster prominence → Notch1 intracellular domain regulation of transcription | 0.66, p=0.007 | | x | x |
| M7 | Laplace of Gaussian intensity entropy → RNA polymerase III transcription | 0.62, p=0.031 | | x | |
| M8 | Laplace of Gaussian intensity standard deviation → Pyruvate metabolism and citric acid TCA cycle | 0.72, p=6e-5 | | x | |
| M9 | Wavelet intensity variance → Trafficking of GLUR2 containing AMPA receptors | 0.64, p=0.020 | x | | |
| M10 | Shape compactness and shape sphericity → TRAF6 mediated NFkB activation | 0.66, p=0.003 | | x | x |
| M11 | Wavelet texture cluster tendency → Platelet aggregation plug formation | 0.69, p=6e-4 | | x | x |
| M12 | Wavelet texture entropy → G0 and early G1 | 0.65, p=0.007 | x | x | x |
| M13 | Laplace of Gaussian intensity entropy → RNA polymerase II transcription initiation and promoter opening | 0.68, p=0.001 | | x | |

**Source data 1.** Radiomic pathway predictors.

## Prognostic value of radiomic signatures

To build on previously published results, we investigated prognostic value of an existing radiomic signature for survival of lung cancer. We fitted a Cox proportional-hazards model of this signature on Dataset1 and observed significant validation by the concordance-index (CI) on Dataset2 (CI = 0.60, Noether p=0.04). Furthermore, we tested combinations of clinical, genetic, and radiomic data and observed that the combinations of data types tended to result in higher performances than given by the individual data alone (*Figure 5*). In particular, the performance of a clinical model increased from CI = 0.65 (Noether p=0.001) to CI = 0.73 (p=$2\times10^{-9}$) when adding the radiomic and an existing gene signature (*Hou et al., 2010*); this increase was significant at p=0.001 by permutation test. This combined radiomic-genetic-clinical model also performed significantly better than the combined radiomic-clinical model (p=0.007) and the clinical-genetic model (p=0.01). Adding radiomics to clinical data alone did not result in a significant increase (p=0.3). We repeated this analysis with a novel radiomic survival signature and other published gene signatures (*Yuan et al., 2004*; *Chen et al., 2007*; *Hsu et al., 2009*), and found that the clinical-genetic-radiomic models consistently yielded the highest performances in nearly all cases (*Figure 5—figure supplement 1* and *Figure 5—figure supplement 2*).

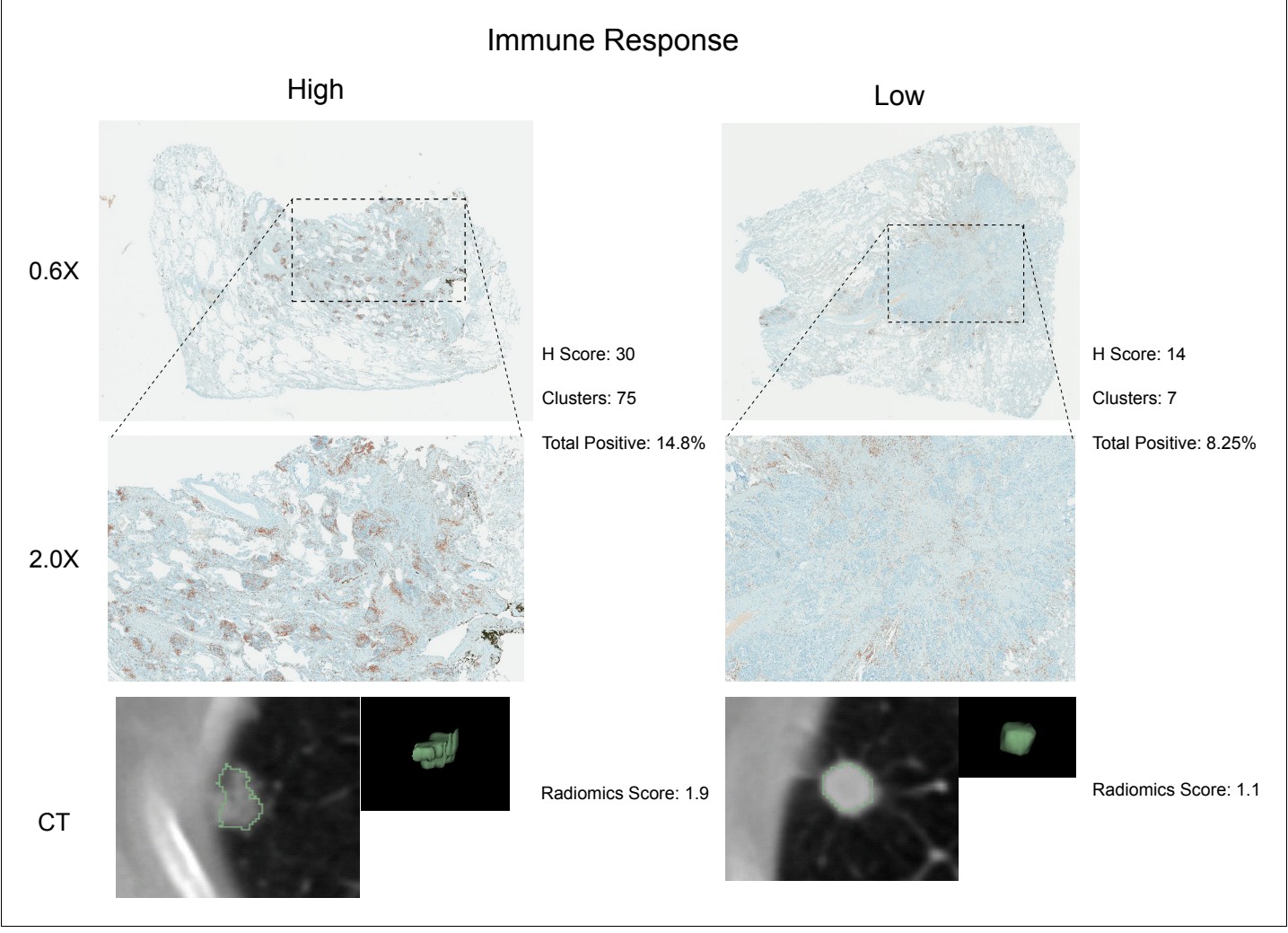

**Figure 4.** Test for agreement between radiomic and pathological immune response assessment. Two representative cases are shown where radiomic predictions of immune response were confirmed by immunohistochemical staining for nuclear CD3 highlighting lymphocytes in brown. Each case is displayed in 0.6X and 2.0X magnification of the tumor slides, and an axial slice of the corresponding diagnostic CT scan and the total tumor volume is given for comparison. Automated quantifications of lymphocytes are displayed in addition to the radiomics score incorporated to classify into high and low responders.

The following figure supplement is available for figure 4:

**Figure supplement 1.** Representative cases of immunohistochemical staining for RelA.

## Discussion

Medical imaging plays a crucial role in cancer diagnosis, treatment, and response monitoring. Radiomics allows quantification of the radiographic phenotype of a tumor (*Kuo and Jamshidi, 2014*; *Gillies et al., 2010*; *Rutman and Kuo, 2009*), but the underlying connections of radiomics to tumor biology and clinical factors have not been elucidated yet. In this study, we identified novel and consistent associations between radiomic phenotype data, underlying molecular pathways, and clinical factors of patients with lung cancer in a North American cohort, and validated our findings in a European cohort and with immunohistochemical staining. In addition, we presented radiomic predictors for pathway activations, and demonstrated the complementary prognostic value of combining radiomic, genetic, and clinical information.

Preliminary studies have previously investigated associations between imaging features, clinical factors, and molecular data for a number of cancer types as outlined in recent reviews (*Gillies et al.,*

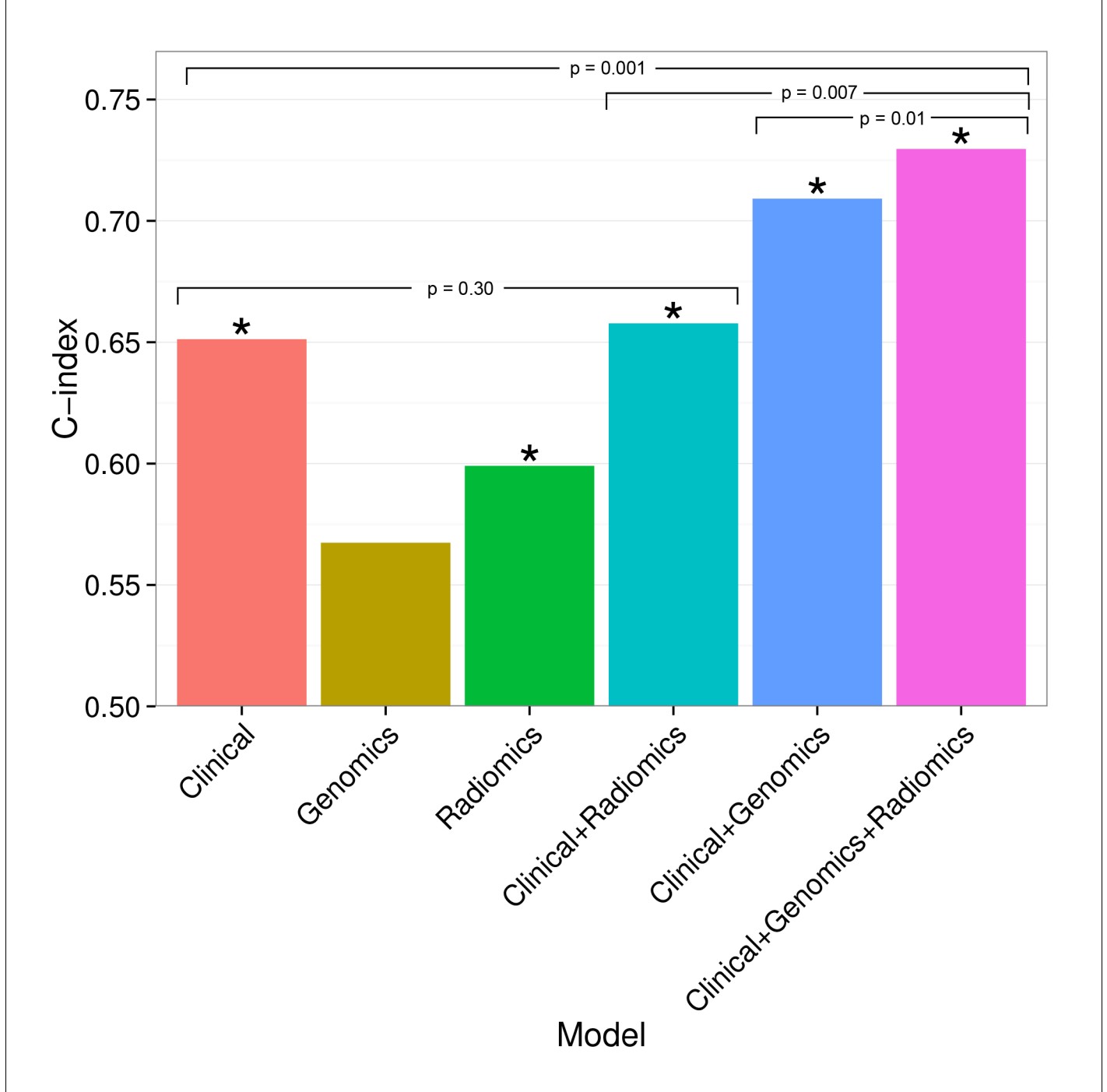

**Figure 5.** Combining prognostic signatures for overall survival. We tested combinations of clinical, genomic, and radiomic signatures. To a clinical Cox proportional-hazards regression model with stage and histology, we first added a published gene signature and next a published radiomic signature. These models were fitted on Dataset1 and evaluated with the C-index (CI) on Dataset2. An asterisk indicates significance ($p<0.05$). Combining different data types resulted in increased prognostic performances. By adding radiomic and genomic information, the initial performance of the clinical model was increased from CI = 0.65 (Noether p=0.001) to CI = 0.73 (p=$2\times10^{-9}$).

The following figure supplements are available for figure 5:

**Figure supplement 1.** Prognostic performance of two radiomic signatures (i.e., a previously published and a novel signature) combined with genetic and clinical information.

*Figure 5 continued on next page*

Figure 5 continued

**Figure supplement 2.** Prognostic performance of two radiomic signatures combined with different gene signatures and clinical information.

*2016*; *Kuo and Jamshidi, 2014*; *Gillies et al., 2010*; *Rutman and Kuo, 2009*; *Cook et al., 2013*). Our analysis builds on these studies in that we performed a rigorous classification of a comprehensive set of radiomic features in terms of underlying molecular pathways on a genome-wide scale and clinical factors in large and independent cohorts. Although the long-term vision is to augment clinical decision making, the current goal of our study is to satisfy the need of the radiomic and oncological community to better understand the underlying biological rationale of radiomic predictions. Furthermore, we are the first to publicly share all study data and analysis code with the growing radiomic and biomedical community to enable further translational research.

We identified and independently validated thirteen radiomic-pathway modules with coherent expression patterns, eleven of which were significantly associated with OS, stage, or histology. By basing these clinical associations exclusively on radiomic features, we could demonstrate that the associated molecular pathways robustly matched radiomic- based hypotheses. For example, based on radiomic features modules M2, M9, and M12 were prognostic and also associated with stage. These modules were highly enriched for immune system, p53, and cell-cycle regulation pathways, biological processes that are widely recognized to play key roles in lung cancer. For example, it has been established that cell cycle regulation is of utmost importance in lung cancer (*Baldi et al., 2011*). Furthermore, the status of p53 is reported to be a predictor of survival in lung cancer patients (*Ahrendt et al., 2003*) and a recent review has laid out how p53 can modulate innate immune system responses (*Menendez et al., 2013*). Radiomic features in these prognostic modules M2, M9, and M12 quantified textural entropy and dispersion image intensity values suggesting associations between textural heterogeneity, cell cycling, and prognosis. Therefore, these results suggest that noninvasive radiomic surrogates may benefit diagnostic methods in assessing cell cycling and immune system states of tumors.

We aimed at confirming our statistical results indicating connections between radiomics, immune response, and survival by immunohistochemical staining of lymphocytes in cases for which a relatively high or low immune response was predicted according to a radiomics score. We generally found high agreement between pathology and radiomics, especially in cases where immune response was predicted to be high. In cases of predicted low responders that showed high pathological immune response, the cause of disagreement may be a heavy distribution of CD3 clusters in the extreme periphery of the tumor with very little staining in the bulk of the tumor. In cases of predicted high responders that showed little to no immune response, this could be due to the lack of normal tissue margin around the edge of the tumor section or a sampling effect. Similarly, we stained tissue for RelA, the p65 subunit of NFkB, to validate radiomic predictions of inflammation. Overall, we found high agreement between pathology and radiomics, although at lower statistical significance. Future studies with whole mount sections stained with multi-plex phenotyping can help determine the relationship between a radiomic and a genetic immune or inflammation signature, and the gold standard.

A variety of textural features were also associated with stage and histology (module M5). Similar associations have been reported by *Ganeshan et al. (2010)*, who suggested that 2D texture features of lung cancer CT scans could predict if tumor stage was II or above. Here, we found that texture features were enriched for axon guidance and lipoprotein metabolism. Furthermore, we observed strong associations between image intensity entropy features and pathways involved in gene expression, transcription regulation, and mitochondrial processes (M13 and M7). Previous research has suggested that imaging can detect consequences of an increase in the hypoxia-inducible factor as a result of absence of oxygen (*Gillies et al., 2010*). Hence, if extracting quantitative information about mitochondrial pathways from medical images leads to assessment of hypoxia status of a tumor, this may ultimately aid in clinical decision-making as alternative therapies for hypoxic tumor areas are being developed (*Denny, 2010*; *Bryant et al., 2014*). Indeed, previous work has indicated that CT pixel intensities correlate with hypoxia markers such as Glut-1 and pimonidazole (*Ganeshan et al., 2013*). Those two modules (M1 and M3) that were not associated with any of the tested clinical factors were relatively small modules; these modules suggested radiomic associations to circadian clock and neural system. The impact of these pathways to the clinical factors we tested

is not apparent from current lung cancer literature, which could explain why these modules did not show clinical associations.

Our results further suggest that radiomic approaches could have the potential to predict molecular states of pathways. We found that the highest predictors of every module was also a suitable representative of the overall biological and radiomic themes of that module. Amongst these examples of pathways that showed high predictability in terms of radiomics, we found various pathways essential for tumorigenesis such as cell cycle pathways (e.g., G0 and early G1), signaling pathways (e.g., Notch and NfKB), and tumor suppressor pathways (e.g., COP1 autodegration and p53). Furthermore, we tested those radiomic pathway predictors for predictive value of driver mutations. Thereby, the highest performances were found for mutations in EGFR and KRAS, which is in line with current radiomic-genetic literature (*Aerts et al., 2016*; *Gutman et al., 2015*; *Liu et al., 2016*; *Rizzo et al., 2016*). Interestingly, however, the highest performance for the tumor suppressor and cell cycle regulator TP53 we found was given by a textural entropy feature that also predicted G0 and early G1 (module M12). In addition, features expressed selectivity for predicting mutations, which was suggested previously (*Gutman et al., 2015*). These results highlight the diagnostic potential, as ready information on pathway and mutation status may permit advanced patient stratification. Previous studies have indicated that gene expression can be predicted by imaging features (*Gevaert et al., 2012*; *Segal et al., 2007*; *Gevaert et al., 2014*). To our knowledge, however, no study has examined and independently validated radiomic models for specific pathways, including biological validation such as immunohistochemical staining.

Finally, we verified a previously described prognostic radiomic signature and observed that the best performance is achieved when combining radiomic, genetic, and clinical data. These results strongly suggest that radiomic data contain complementary prognostic information and are robust, as the published radiomic signature (*Aerts et al., 2014*) has not been tested on our data before. Notably, these prognostic improvements were relatively stable to substitution of radiomic or gene signatures. A related indication of improved survival predictions by combining imaging features and molecular data has been recently given for glioblastoma, however without validation (*Nicolasjilwan et al., 2015*). It is worth noting that for the first time we also demonstrate that radiomic prognostication generalizes across cohorts from different continents.

Three research tracks have recently been proposed for clinical translation of such imaging biomarkers (*O'Connor et al., 2017*), including biological validation, technical validation, and evaluation of cost-effectiveness. Our study conforms with several of these roadmap recommendations by advancing results on a previously proposed radiomic signature (*Aerts et al., 2014*) with additional biological validation and investigations on how genetic data and clinical factors impact this signature. Fixing a radiomic signature for technical validation and cost-effectiveness verification should be considered in subsequent studies to overcome additional translational gaps. Although the long-term vision would be to augment clinical decision making, the current goal of our study is to contribute in satisfying the need of the radiomic and oncological community to understand the underlying biology of radiomic predictions.

Our study is limited by its retrospective nature. Imaging protocols are not standardized and hence variability in CT acquisition and reconstruction parameters is inherent in clinical practice. However, despite this, no corrections by cohort or scanner type were made in this study illustrating the translational aspect of our results that generalized across institutions. Hence, we expect that the performance of radiomics will further improve, as imaging data are becoming more standardized. In fact, multiple studies have already documented the robustness of radiomic feature extractions in terms of reproducibility and repeatability in test/re-test settings (*Fried et al., 2014*; *Balagurunathan et al., 2014*; *Leijenaar et al., 2013*; *Parmar et al., 2015a*; *Aerts et al., 2014*; *Grove et al., 2015*). Another limitation of this study is that the current cohorts mainly focused on early stage (I - III) tumors, hence generalization of radiomic-genomic associations to late stage tumors should be drawn with precaution only. However, most radiomic applications do focus on early stage tumors as the current radiomic approach requires segmentation of tumors which for late stage tumors remains to be of particular complexity. Furthermore, although our study provides multiple facets of validation, immunohistochemical validation was restricted to considerably smaller sample sizes as compared to our statistical validations due to limited availability of frozen tissue. Prospective protocols can ensure availability of sufficient tissue for additional validation.

Biological material investigated in this study has been acquired by single-needle biopsies, thus the interpretation of our genomic data is limited due to heterogeneity of lung cancer tumors. However, as our results validated in independent data and because known drivers of tumorigenesis were among the main pathways found to be associated with radiomic features, this suggests that these associations have been established in an early evolutionary step in tumorigenesis and are therefore reasonable representatives of the overall tumor. Prospective studies with defined spatial matchings of biopsies and/or single cell analyses could provide deeper insight into whether the strengths of these associations can be further increased. Prospective studies will also be required to assess clinical utility of combining radiomic, genomic, and clinical data into prognostic models.

In conclusion, this study presented novel and consistent associations between radiomics, molecular pathways, and clinical factors. We applied an independent discovery and validation design on large patient cohorts from different continents with enough variability that allowed confidence in the generalization of our results. Furthermore, we performed biological validation and demonstrated that radiomics predicts molecular pathway status and thus improves the prognostic performances of clinical and gene signatures. The clinical impact of our results is illustrated by the fact that it advances the molecular knowledge of automated radiomic characterization of tumors, information currently not used clinically. This may provide opportunities to improve decision-support at low additional cost as imaging is routinely used in clinical practice as standard of care.

## Materials and methods

### Discovery and validation data

Data underlying this study is made publically available with this article; Dataset1 can be downloaded as *Figure 2—source data 1* and Dataset2 can be downloaded as *Figure 2—source data 2*. We analyzed two cohorts of patients with non-small cell lung cancer (NSCLC), Dataset1 and Dataset2, each consisting of pretreatment diagnostic computed tomography (CT) scans, gene expression profiles, and clinical data. While the larger cohort Dataset1 (North American) is novel and served as a discovery cohort, Dataset2 (European) has been previously published with CT scans and gene expression data (*Aerts et al., 2014*), and was used for independent validation of our findings. Patients in Dataset1 were treated in the Thoracic Oncology Program at the H. Lee Moffitt Cancer Center, Tampa, Florida, USA; we included patients with diagnosed primary tumors who underwent surgical resection and collected contrast-enhanced CT scans obtained within 60 days of the diagnosis between years 2006 and 2009. Patients in Dataset2 were treated at MAASTRO clinical, Maastricht, NL; we included patients with confirmed primary tumors who received surgery. Further details of Dataset2 are given by *Aerts et al. (2014)*. The majority of CT scans were recorded to be contrast-enhancing (89% and 71% of patients in Dataset1 and Dataset2, respectively).

For analyses involving CT scans and gene expression data, 262 and 89 patients were available for Dataset1 and Dataset2, respectively. In addition, clinical data were available for 224 and 87 patients, respectively. Clinical outcomes investigated were overall survival (OS), pathologic TNM stage (combined T, N, and M stages, according to the latest version 7 of the IASLC guideline for lung cancer [*Mirsadraee et al., 2012*]), and pathologic histology (grouped into adenocarcinoma, squamous carcinoma, and others). Clinical stage and histology were used when pathologic information was not available. Tumors in these cohorts were mainly early stage; in Dataset1 among the 224 clinically annotated cases 26 were stage IIIB or IV and in Dataset2 among the 87 clinically annotated cases 3 cases were stage IV. These late stages have been grouped into 'other' for analysis. Further clinical cohort characteristics are given in *Table 1*.

For tumors in both cohorts, expression of 60,607 probes was measured on a custom Rosetta/ Merck Affymetrix 2.0 microarray chipset (HuRSTA_2a520709.CDF, GEO accession number GPL15048) by the Moffitt Cancer Center. Gene expression of Dataset2 is available also at Gene Expression Omnibus (GEO) through accession number GSE58661. Gene expression values were normalized with the robust multi-array average (RMA) algorithm (*Irizarry et al., 2003*) implemented in the 'affy' Bioconductor package (*Gautier et al., 2004*). Probes have been curated by choosing the most variant representative among probes mapping to the same gene identifier (Entrez Gene) resulting in a total of 21,766 unique genes.

## Radiomic features

We extracted 636 features grouped into I) tumor intensity (voxel statistics), II) shape, III) texture, IV) wavelet, and V) Laplace of Gaussian features. Group I-IV features have been defined as specified by *Aerts et al. (2014)*. In addition, we added new features to Group III (see GLSZM below). *Group I* features are first-order statistics (e.g. mean, skewness) of all voxel intensity values in the tumor volume mask. *Group II* features describe the shape and size of a tumor (e.g. compactness). *Group III* features quantify texture in tumor images describing clustering of voxels with similar appearance by means of a gray-level co-occurrence matrix (GLCM), a run-length gray-level matrix (RLGL), or a gray-level size-zone matrix (GLSZM). These features quantify how frequent voxels of same gray-level are adjacent to each other (GLCM), how many voxels of the same gray-level appear in a consecutive run (RLGL), or the sizes of flat zones, areas of same gray-level in all directions (GLSZM). *Group IV* features are Group I-III features (except GLSZM) assessed after a wavelet decomposition of the image, which highlights sharp transitions in the intensity frequency spectrum. *Group V* consists of Group I features that have been calculated after applying a Laplace of Gaussian transformation to the image, which highlights edge structures. Detailed description and analytical definitions of the features added to the *Aerts et al. (2014)* feature set (n = 440) are given in *Supplementary file 1*. Features were calculated in 3D. For normalization, slice thicknesses of all scans were interpolated to a voxel sizes of $1 \times 1 \times 1$ mm$^3$.

## Pathway analysis

To test if a radiomic feature was associated with a molecular pathway, Spearman's rank correlation coefficient *rho* was calculated for the expression of every gene across all patients and weighted by -log10(*p*), where *p* is the p-value of *rho*. The resulting gene rank was input to a preranked gene set enrichment analysis (GSEA) algorithm (*Subramanian et al., 2005*) version 2.0.14 on the C2 collection version 4 of the Molecular Signature Database (MSigDB) (*Liberzon et al., 2011*). This collection contains the expert-curated set of pathways from the Reactome database (*Joshi-Tope et al., 2005*). Those 511 out of 674 pathways were considered that contained at least 15 and at most 500 genes. GSEA reports normalized enrichment scores (NESs) for every pathway, which we further analyzed.

## Radiomic-pathway association modules

To identify coherently expressed expressed features and pathways, a matrix holding an NESs for every pair of radiomic feature and Reactome pathway was biclustered with the Iterative Signature Algorithm (ISA) using the 'isa2' and 'eisa' packages in R and Bioconductor (*Bergmann et al., 2003*; *Csárdi et al., 2010*). As a result, each bicluster contains a set of coherently expressed features and pathways and is referred to as module. Potential module redundancy was limited using the 'isa.unique' function in the 'isa2' package with a maximum correlation threshold of 0.3. To avoid parameter sensitivity with ISA, row and column clustering seed thresholds were set to a liberal sequence of 1.5 to 2.5 by 0.5 to include all potential signals. This procedure yielded 20 putative modules. To validate these modules, we developed and applied a correlation based statistic $r := mean(C_X) + mean(C_Y)$, where $C_X$ and $C_Y$ are the Spearman rank correlations of all pairs of features and pathways in a module, respectively. The true $r$ was calculated for every module in Dataset1 and validated on Dataset2 with random permutation tests (N = 1000). After correcting for multiple-hypothesis testing with the false-discovery-rate (FDR) (*Benjamini and Hochberg, 1995*), the validation resulted in 13 significantly enriched modules (FDR < 0.05). In total, the modules captured the associations between 210 radiomic features and 206 pathways.

Module size was defined as $n/N + m/M$, where $n$ and $m$ are the number of features and pathways in a module, respectively, and $N = 636$ and $M = 511$ are the total numbers of features and pathways across all modules, respectively. Overlap of two modules was defined by the Jaccard index (*Theodoridis and Koutroumbas, 2008*), which is the size of union of features divided by the size of intersection of features of two module. Hereby, same feature names under different transformations were considered equivalent.

## Pathway predictions

To test radiomic pathway predictors, we used gene set variation analysis (GSVA) in Bioconductor (*Hänzelmann et al., 2013*) to calculate pathway enrichment scores per patient. Next, we fitted univariate logistic regression models of every feature to predict the NES sign of pathways (which

corresponded to activation or deletion) in Dataset1. We assessed the concordance between the predicted probabilities of the pathway sign and the true sign with the area under the curve (AUC) of the receiver operator characteristic (ROC) (*Bradley, 1997*). The strongest predictor of each module according to the AUCs in Dataset1 was evaluated on Dataset2 for validation; significance of AUCs was calculated according to Noether for binary outcomes (*Pencina and D'Agostino, 2004*).

## Associations to clinical factors

Associations to OS were assessed by calculating the mean concordance-index (*Harrell et al., 1982*) of all features in a module univariately using the 'survcomp' package in Bioconductor (*Schröder et al., 2011*), and by validating this statistic with repeated random permutation tests (N = 1000). Similarly, associations to stage and histology were assessed by the mean of Kruskal-Wallis chi square statistics and permutation tests. As clinical information was not part of the module identification process, a meta-analysis of the results in Dataset1 and Dataset2 was conducted to account for sample size differences and other dataset specific variations. For this, a Fisher Z-transformation (*Whitlock, 2005*) of the independent p-values in both datasets was employed for every module with weights equal to the respective sample sizes in Dataset1 and Dataset2.

We tested additive prognostic effects of integrating radiomic, gene expression, and clinical data by combining in a Cox proportional-hazards model the predictions of (I) a clinical Cox model with stage and histology, (II) an NSCLC OS gene signature, and (III) an NSCLC OS radiomic signature. We tested five published gene signatures (*Hou et al., 2010*; *Yuan et al., 2004*; *Chen et al., 2007*; *Hsu et al., 2009*) without inclusion of clinical and radiomic data and retained the strongest performing signature by *Hou et al. (2010)* to challenge potential performance increases. To test for generalizability of radiomics, we tested a published radiomic signature by *Aerts et al. (2014)* and a novel signature developed in the current study. We developed this novel radiomic signature using a supervised feature selection algorithm followed by a stepwise Cox regression approach on Dataset1: First, we employed the minimum-redundancy maximum-relevance (mRMR) algorithm implemented in the 'mRMRe' R package (*De Jay et al., 2013*) on all radiomic features with respect to OS to select a non-redundant, highly informative ranked set of complementary features. Next, we trained Cox models incrementally, adding features starting by the highest ranked feature. We performed repeated random cross-validation (N = 1,000) to measure the performance of each model, and retained the model with the highest mean CI. Finally, these fitted models were tested on Dataset2 for validation.

All statistical analyses were carried out using the R software (*R Development Core Team, 2013*) version 3.1.0 on a Linux operating system. Details of version numbers of utilized packages are available in *Supplementary file 2*.

## Immunohistochemical staining for CD3

We selected 25 cases each that were predicted to have high and low immune response by using the value of the radiomic feature in the prognostic modules M2, M9, and M12 that showed the highest absolute correlation to the mean expression of genes in the CTLA4 inhibitory pathway that is supported to be associated with immune activity (*Postow et al., 2015*; *Pardoll, 2012*; *Wolchok and Saenger, 2008*). In total, 22 cases were available with enough tumor tissue and sufficient staining quality. Tumor cross section slides were stained using a Ventana Discovery XT automated system (Ventana Medical Systems, Tucson, AZ) as per manufacturer's protocol with recommended reagents. Briefly, slides were deparaffinized with EZ Prep solution (Ventana) and a heat-induced antigen retrieval method was used under mild cell conditioning using CC1 antigen retrieval buffer (Ventana). A rabbit primary antibody for CD3, (790–4341, Ventana) was used at supplied concentration and incubated for 16 min. Next a Ventana OmniMap Anti-Rabbit Secondary Antibody was applied to the samples for 16 min and the Ventana ChromoMap kit was used as the detection system. Slides were then counterstained with Hematoxylin and dehydrated. Finally, the slides were cover slipped as per normal laboratory protocol.

## Immunohistochemical staining for RelA

We selected 25 cases each that were predicted to have high and low NFkB activity. The same procedure as for the CD3 staining was applied, with the exception that a standard cell conditioning was used with CC2 antigen retrieval buffer (Ventana). Furthermore, a rabbit polyclonal primary antibody

for RelA (NFkB p65), (Spring Biosciences E2750) was used at 1:600 dilution* and incubated for 32 min. In total, 24 cases were available with enough tumor tissue and sufficient staining quality.

## Evaluation a immunostained slide

The lymphocytes are highlighted by brown nuclear staining of CD3. The staining pattern was analyzed by a board-certified pathologist (MB) and scored into low and high enrichment. The percentage and intensity (weak 1+, moderate 2+ and intense 3+) of staining were recorded as well as the number and size of clustering of CD3 positive cells. The pathologist also chose the appropriate area from each sample for image analysis. We observed that the tissue section that has a complete cross section of the tumor with a complete rim of adjacent benign lung parenchyma is most ideal for image analysis. This is because the lymphocytic infiltration is commonly present at the periphery of the tumor. In addition to this assessment by a pathologist, a computational system was implemented for automatic evaluation (*Supplementary file 3*).

## Acknowledgements

We thank Yuhua Gu and Alberto Garcia for their help with curating and segmenting patient CT images for Dataset1. We would also like to thank Joseph Johnson, Jonathan V Nguyen, Michelle Fournier and Jeanette M Rheinhardt for their assistance in the analysis of stained lymphocytes. Finally, the authors thank the Enterprise Research Infrastructure & Services at Partners Healthcare for provision of their HPC infrastructure. This work has been supported in part by the Tissue Core Facility at the H Lee Moffitt Cancer Center & Research Institute, an NCI designated Comprehensive Cancer Center (P30-CA076292).

## Additional information

### Competing interests

RJG: declares a collaboration with HealthMyne. The other authors declare that no competing interests exist.

### Funding

| Funder | Grant reference number | Author |
| --- | --- | --- |
| National Institutes of Health | NIH-USA U24CA194354 | Hugo JWL Aerts |
| National Institutes of Health | NIH-USA U01CA190234 | Hugo JWL Aerts |
| National Institutes of Health | NIH/NCI U01CA143062 | Robert J Gillies |
| National Institutes of Health | NIH/NCI P50CA119997 | Robert J Gillies |
| QuIC-ConCePT | IMI JU Grant No. 115151 | Philippe Lambin |
| Technologiestichting STW | 10696 DuCA | Philippe Lambin |
| Dutch Cancer Society | KWF UM 2009-4454 | Philippe Lambin |
| Dutch Cancer Society | KWF MAC 2013-6425 | Philippe Lambin |
| Gattuso Slaight Personalized Cancer Medicine Fund | | Benjamin Haibe-Kains |

The funders had no role in study design, data collection and interpretation, or the decision to submit the work for publication.

### Author contributions

PG, Conceptualization, Resources, Data curation, Software, Formal analysis, Validation, Investigation, Visualization, Methodology, Writing—original draft, Project administration, Writing—review and editing; OS, Resources, Data curation, Supervision, Investigation, Writing—original draft, Project administration, Writing—review and editing; NE-H, Formal analysis, Investigation, Methodology, Writing—review and editing; MMB, Resources, Data curation, Software, Formal analysis, Validation, Investigation, Visualization, Methodology, Writing—review and editing; ERV, CP, RTHL, Resources,

Software, Writing—review and editing; BH-K, Conceptualization, Software, Formal analysis, Supervision, Writing—review and editing; PL, Resources, Data curation, Software, Funding acquisition, Project administration, Writing—review and editing; RJG, Conceptualization, Resources, Data curation, Supervision, Funding acquisition, Investigation, Writing—original draft, Project administration, Writing—review and editing; HJWLA, Conceptualization, Resources, Data curation, Software, Supervision, Funding acquisition, Investigation, Methodology, Writing—original draft, Project administration, Writing—review and editing

### Author ORCIDs
Patrick Grossmann, http://orcid.org/0000-0003-4918-6902
Robert J Gillies, http://orcid.org/0000-0002-8888-7747
Hugo JWL Aerts, http://orcid.org/0000-0002-2122-2003

### Ethics

Human subjects: The University of South Florida institutional review board approved and waived the informed consent requirement (IRB # 16069) retrospective study of Dataset1; data were collected and handled in accordance with the Health Insurance Portability and Accountability Act. Informed consent for gene expression collection was written and oral. For acquisition of imaging and clinical data USF IRB approved protocol (108426) provided a waiver of informed consent for this retrospective study. Data collection and analysis of Dataset2 was carried out in accordance with Dutch law; the corresponding institutional review board approved the study. All patient data were anonymized and de-identified prior to the analyses.

## Additional files

### Supplementary files

• Supplementary file 1. Radiomic feature definition and further description towards meaning of feature groups.

• Source code 1. Analysis code. Source code used to analyse data and generate figures.

• Supplementary file 2. Exact p-values of modules and list of used R packages and their versions used for analysis.

• Supplementary file 3. Methods for automated pathological call assessment.

### Major datasets

The following previously published dataset was used:

| Author(s) | Year | Dataset title | Dataset URL | Database, license, and accessibility information |
|---|---|---|---|---|
| Aerts HJ, Grossmann P | 2014 | 89 NSCLC patients with gene expression profiles and matching CT imaging data available at TCIA | https://www.ncbi.nlm.nih.gov/geo/query/acc.cgi?acc=GSE58661 | Publicly available at the NCBI Gene Expression Omnibus (accession no: GSE58661) |

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
