## [Decision Letter]

Thank you for submitting your article "Identification of Molecular Phenotypes in Lung Cancer by Integrating Radiomics and Genomics" for consideration by *eLife*. Your article has been favorably evaluated by Aviv Regev (Senior Editor) and three reviewers, one of whom, Trever Bivona (Reviewer #1) who served as Guest Editor. The following individual involved in review of your submission has agreed to reveal their identity: James O'Connor (Reviewer #3).

The reviewers have discussed the reviews with one another and the Reviewing Editor has drafted this decision to help you prepare a revised submission.

Summary:

This study presents of series of medical imaging based findings that link radiographic tumor features to clinical and molecular features in lung cancer (via radiomics). This radiomic approach was used in combination with clinical and genetic characteristics of the tumors to generate prognostic assessments, with improved performance. Overall, this study presents interesting insights and suggests the utility of radiomics for informing the understanding of the pathogenesis of lung cancer.

Several issues should be addressed to confirm the validity and clarity of the data presented and better link the findings to the current molecular state of the lung cancer field.

Essential revisions:

1) The cohort analyzed by the authors is mostly early stage lung cancer. Were any stage IIIB or IV tumors included? This stage bias should be noted in the text and the findings contextualized accordingly.

2) It is important to at least annotate the tumors studied for major driver mutations in lung cancer (e.g. mutant EGFR, mutant KRAS, ALK gene rearrangements, mutant BRAF, mutant NF1, TP53, and so on). Associating radiomic features with these genotypes that direct clinical management would greatly enhance the relevance and impact of the current study.

3) There should be better molecular validation by biomarker staining in the tumor tissue of the gene sets that are used to score different molecular phenotypes. As it stands, it is unclear how and why the gene sets were selected for classification of the tumors and whether such in silico based classification translates into the human tissue in a meaningful way at the protein level (of key pathway biomarkers). An example is to perform RelA IHC to validate the NFkB finding in the relevant tumors.

4) Were the 5 or so radiomic values "aggregated" into a single index per tumor to perform the clustering analysis with the gene set signature? If not, more elaboration as to how the clustering analysis was done would be very helpful.

5) Please clarify the purpose and message of the network presented in Figure 3. First, what does it mean for a module to correlate with the respective clinical feature. For instance, are tumors in modules 2, 9, and 12 derived from patients with a lower OS compared to the rest of the cohort? If so, perhaps the authors could comment on the importance of defining this, either in the development of their methodology or to clinical decision making. Moreover, could the authors elaborate as to why some modules do not correlate with any clinical feature.

6) It is appreciated that the modules were verified with an additional patient data set independent of the discovery data set, but this study begs for validation of pathway activation in some of the tissues directly using immunohistochemistry. Modules 4, 6, and 8 seem like low hanging fruit given the enrichment of well characterized signaling pathways with an abundance of markers of pathway activation for IHC.

7) Related to point #6, is it not troubling that the IHC for CD3 did not appear to agree with the prediction of high or low immune response for modules 2, 9, and 12?

8) Can the authors elaborate on the clinical significance of developing a prognostic marker for a cancer that is overall highly aggressive? How will the prognostic power of the combination of R + G + C impact clinical decision making?

9) What is the rationale for extending the 440 features in the previous Nat Commun study to 636 features? Please comment on this specific point for this study, but also for the subsequent translation of the radiomics idea could the authors discuss how to move from deriving and testing one or another 'radiomic signature' to deciding on a final signature and then locking this down, for future application. Is this approach planned, would it be tumor-therapy specific etc.? Some discussion of this is important in biomarker translation. The imaging biomarker roadmap in Nat Rev Clin Oncol 2017 should be cited in this regard.

10) Since this study has several positives (use of test-validation, multisite data, advanced tailored statistics, use of pathology and outcome data etc.) it would be important to a) state how this study conforms with several of the roadmap recommendations and b) explicitly highlight how the biomarker = radiomics signature family (the one here is different from that in the previous paper in Nat Commun) has been advanced in this study down the biological/clinical validation pathway. This is alluded to, to some extent in the eighth paragraph of the Discussion, but this point could be strengthened.

11) The section 'Immunohistochemical investigation of radiomic immune response assessment' seems less convincing that the data from other sections as there is N=15. Is this justified as an analysis based on the very small N? Please justify the inclusion of these data for this section.

12) Many of the statistical associations are strong. But it is difficult to understand how the radiomic metrics relate to biological pathways that clearly measure something quite different (scale, biology). Is this really going to convince oncologists that a radiomic metric (e.g. a gray scale texture) is an adequate measure of say cell cycling? Does this seem plausible and if not then is that a translational barrier? Please discuss this a bit more in the text, as I think this will be seized upon by readers over the next few years.

---

## [Author Response]

[…] Several issues should be addressed to confirm the validity and clarity of the data presented and better link the findings to the current molecular state of the lung cancer field.

In addition to the responses below, to better reflect the specific focus of study we would like to change the current title of the manuscript into: “Defining the biologic basis of radiomic phenotypes in lung cancer”.

Essential revisions:

1) The cohort analyzed by the authors is mostly early stage lung cancer. Were any stage IIIB or IV tumors included? This stage bias should be noted in the text and the findings contextualized accordingly.

We thank the reviewer for the time in critically assessing our study. Indeed, our study cohorts contain mainly early stage tumors, although several stage IIIB and IV cases are included; specifically 26 in the discovery cohort. In the radiomic analysis of histology, these cases were grouped into “Other” to avoid imbalanced outcome groups. In our revised manuscript, we emphasized this important point in the Materials and methods and Discussion. Furthermore, a relevant issue with these late stage tumors is that they are difficult to segment, and are thus a challenge for radiomics of lung lesions. The approach that is currently being taken is to extract radiomic features from metastatic sites, e.g. liver, lymph nodes, adrenals; but this is quite distinct from the focus of the current work. Our text has now also been updated to reflect this, as well.

2) It is important to at least annotate the tumors studied for major driver mutations in lung cancer (e.g. mutant EGFR, mutant KRAS, ALK gene rearrangements, mutant BRAF, mutant NF1, TP53, and so on). Associating radiomic features with these genotypes that direct clinical management would greatly enhance the relevance and impact of the current study.

We thank the referee for this insightful comment. We agree that associating radiomic features with widely established lung driver mutations would enhance our study. Therefore, we retrospectively queried our institutional database to include all available sequencing records. For our revised manuscript, we can now present additional data on 60 patients whose primary tumors were profiled with Sanger sequencing. For these patients we analyzed how the representative features of every module of our analysis predict mutant EGFR, KRAS, and TP53. We found strong predictive value for EGFR and KRAS, but only moderate predictive value for TP53. In particular, features that predicted EGFR and KRAS were first-order statistics of the voxel intensities and textural features, complementing our previous associations to cell cycling and proliferation pathways. We included these new results in the revised manuscript. The mutation status of the current cohort will be made available in the revised text and shared publicly with all other data. Furthermore, we referenced prior work from our research groups and others wherein radiomic features were associated with EGFR, KRAS, and other lung cancer driving mutations according to the popular model of clonal evolution.

3) There should be better molecular validation by biomarker staining in the tumor tissue of the gene sets that are used to score different molecular phenotypes. As it stands, it is unclear how and why the gene sets were selected for classification of the tumors and whether such in silico based classification translates into the human tissue in a meaningful way at the protein level (of key pathway biomarkers). An example is to perform RelA IHC to validate the NFkB finding in the relevant tumors.

We would like to thank the referee for the apparent interest in our study. Indeed, additional molecular validation of our computationally derived and, on a large and independent cohort, statistically validated results is a positive suggestion. Hence, we selected tumors that were predicted to have the highest and lowest activation of NFkB by shape sphericity, the representative feature of module 10 that was also associated with NFkB activation. Following the suggestion from the reviewer we stained available tissue from these tumors for RelA and assessed the enrichment by an experienced pathologist (author Dr. MB, MD, PhD). Comparison of radiomic and pathological assessment revealed that radiomic feature values were higher for those cases that were also pathologically scored to be relatively higher enriched for RelA; according to a Wilcoxon rank sum test this was at a significance level of *p* = 0.06. We added these new results to our revised manuscript and included a corresponding discussion of these updates.

4) Were the 5 or so radiomic values "aggregated" into a single index per tumor to perform the clustering analysis with the gene set signature? If not, more elaboration as to how the clustering analysis was done would be very helpful.

We thank the referee for this request and apologize if clarity in our Materials and methods was limited. We leveraged a pre-ranked gene-set enrichment analysis using a correlation rank between every radiomic feature and the expression of every gene. This allowed us to calculate a normalized enrichment score for every feature and gene set (i.e., a large matrix), which was the basis for our clustering analysis. In this sense, indeed, we associate every feature with every gene set independently. We updated the text in our revised manuscript to increase clarity of this point.

5) Please clarify the purpose and message of the network presented in Figure 3. First, what does it mean for a module to correlate with the respective clinical feature. For instance, are tumors in modules 2, 9, and 12 derived from patients with a lower OS compared to the rest of the cohort? If so, perhaps the authors could comment on the importance of defining this, either in the development of their methodology or to clinical decision making. Moreover, could the authors elaborate as to why some modules do not correlate with any clinical feature.

We appreciate and thank for the detailed review of our manuscript. Figure 3 summarizes the clinical associations that we found and validated for every module. Specifically, every module describes an association of a cluster of radiomic features and molecular pathways. We tested whether the radiomic features of a module were also associated with overall survival, tumor stage, and histology by C-index and chi-square based statistics as described in our Materials and methods in section “Associations to clinical factors.” With this imaging-based (i.e., genomic-independent) analysis we aimed at verifying whether radiomic-clinical associations of a module also follow the radiomic-genomic associations of that module that are supported by current lung cancer biological literature and hence contribute in closing the gaps of understanding the ‘association-triangle’ of imaging, genomics, and clinical factors. We updated our Discussion regarding why certain modules may not been associated to certain clinical factors.

6) It is appreciated that the modules were verified with an additional patient data set independent of the discovery data set, but this study begs for validation of pathway activation in some of the tissues directly using immunohistochemistry. Modules 4, 6, and 8 seem like low hanging fruit given the enrichment of well characterized signaling pathways with an abundance of markers of pathway activation for IHC.

We thank this reviewer for noting our stringent statistical study design and for suggesting additional staining experiments, which is in line with similar comments. To address these comments, we present additional data on immunohistochemical staining for RelA (a p65 NKkB subunit) to validate radiomic associations with inflammation, and also present additional data on staining for CD3 to validate radiomic associations to immune response. We kindly refer to the corresponding sections of our revised manuscript and hope that the significance of these updates is noted.

7) Related to point #6, is it not troubling that the IHC for CD3 did not appear to agree with the prediction of high or low immune response for modules 2, 9, and 12?

We would like to thank the referee for taking time to assess the results of our manuscript, and apologize for any confusion. We did find pathological agreement for the association with immune response in modules 2, 9, and 12. We used modules 2, 9, and 12 to select the radiomic feature that representing the immune response score. We used this score to select cases that this radiomic feature predicted to show relatively high or low immune response. For these cases, we performed immunohistochemical staining for CD3 (as surrogate for immune response) and found that the pathological assessment of immune response agreed with the radiomic score in the majority of stained cases. In our revised manuscript, we present additional data on this to increase statistical power of our analysis. We updated the revised manuscript to highlight this important point.

8) Can the authors elaborate on the clinical significance of developing a prognostic marker for a cancer that is overall highly aggressive? How will the prognostic power of the combination of R + G + C impact clinical decision making?

According to the National Comprehensive Cancer Network (NCCN), an alliance of cancer centers in the United States, there are a number of treatment options for early stage cancers including, inter alia: adjuvant CTx, chemorads, immune Tx with and without RTx/CTx combinations. We propose that radiomic biomarkers can be used within a decision support system when planning therapy. Realizing that therapy choice is made between clinical oncologist and patient preference, more aggressive radiomic signatures may lead to more aggressive therapies. The superior prognostic power of combining radiomics with genomics and clinical data has long been hypothesized, but up to now no study rigorously investigated whether this holds true. The advantages of combining these inherently different data types seem obvious: while genomics quantifies molecular events happening in tumors, radiomics is able to provide a phenotypic quantification. The hope for clinical decision making is that these data types complement each other in prognostication. We included additional elaboration in the discussion on these fundamental points of our and other's work. At this time, we are not proposing that this be used in a CADx system. This important point was added to the Discussion.

9) What is the rationale for extending the 440 features in the previous Nat Commun study to 636 features? Please comment on this specific point for this study, but also for the subsequent translation of the radiomics idea could the authors discuss how to move from deriving and testing one or another 'radiomic signature' to deciding on a final signature and then locking this down, for future application. Is this approach planned, would it be tumor-therapy specific etc.? Some discussion of this is important in biomarker translation. The imaging biomarker roadmap in Nat Rev Clin Oncol 2017 should be cited in this regard.

We thank the referee for commenting on the various feature sets presented in current radiomic research and their translational impact. With regard to the referenced feature set from our previous study published 2014 in Nature Communications, we simply updated the 2014 feature set with a novel textural matrix (i.e., the gray-level size-zone matrix) to derive new radiomic texture features from. Including this additional matrix in our feature extraction resulted in a total of 636 features; we describe this in more detail in our Materials and methods section and provide the mathematical derivation of those new features in the supplemental material. Generally, radiomics is a dynamic field in which feature sets continue to be enhanced which is a major focus of investigation by a number of groups internationally. What we show is that a parsimonious set derived from a large set of features can be used to improve prognostic value, but indeed in near-future studies we aim at fixing a radiomic signature for definitive clinical evaluation. We believe that the most reasonable way to conduct this would be to define such a radiomic signature from a research group, fix its weights at later stages, and have another, independent group evaluate its prognostic performance whilst having an institutional ethics committee oversee and verify double-blindness of this procedure. Unlike biological biomarkers, technical validation can occur at later stages; however, cost-effectiveness is a crucial track in translation that needs to be considered early on. We include an elaborate discussion including the suggested reference to reflect on all of these important points.

10) Since this study has several positives (use of test-validation, multisite data, advanced tailored statistics, use of pathology and outcome data etc.) it would be important to a) state how this study conforms with several of the roadmap recommendations and b) explicitly highlight how the biomarker = radiomics signature family (the one here is different from that in the previous paper in Nat Commun) has been advanced in this study down the biological/clinical validation pathway. This is alluded to, to some extent in the eighth paragraph of the Discussion, but this point could be strengthened.

We thank the referee for this very positive feedback in correctly noting the multiple efforts we undertook to provide a stringent analysis. We agree that in this regard it would be valuable to emphasize how our study can advance the urgently needed development for accurate and robust imaging biomarkers in clinical settings. We built on the signature that we proposed in the referenced study (Aerts et al., Nature Communications 2014) by investigating the prognostic results of combining features of this signature with genetic and clinical data. In addition to this, we developed a novel signature to investigate whether the results of the previous signature hold for further signatures, as well. To convey this message more clearly, we extended the corresponding paragraphs with more elaborate discussions, including future perspectives.

11) The section 'Immunohistochemical investigation of radiomic immune response assessment' seems less convincing that the data from other sections as there is N=15. Is this justified as an analysis based on the very small N? Please justify the inclusion of these data for this section.

We thank the reviewer for the critical assessment of our study. IHC of patient derived tissue samples is limited in clinical practice by availability of frozen tissue. Therefore, we chose a manageable, yet statistically reasonable sample size; initially, we selected N=20 for 10 samples of high radiomic expression and 10 for low radiomic expression. From these N=20 samples, N=15 samples that contained tumor tissue were located by the institutional tissue core, which still provided us with a sufficiently large sample size to calculate a valid *p*-value with a standard Fisher’s exact test. As the *p*-value is a function of the sample size, insufficient sample size would be reflected in this *p*-value.

We agree, however, that even more efforts in analyzing stained tissue would greatly enhance our study. We therefore performed additional staining for CD3 as described in the Materials and methods and now aimed at doubling our initial sample size in the resubmitted manuscript. We also updated our statistical method to a more quantitative one. Indeed, with these updates we observe a considerable increase in statistical power from *p* = 0.13 to *p* = 0.008. We include these updated results in the revised manuscript with a corresponding discussion of the results and methods (including potential sample size limitations), and thank the reviewers for suggesting additional stains of relevant tumors.

12) Many of the statistical associations are strong. But it is difficult to understand how the radiomic metrics relate to biological pathways that clearly measure something quite different (scale, biology). Is this really going to convince oncologists that a radiomic metric (e.g. a gray scale texture) is an adequate measure of say cell cycling? Does this seem plausible and if not then is that a translational barrier? Please discuss this a bit more in the text, as I think this will be seized upon by readers over the next few years.

We would like to thank the referee for this positive remark and agree that this important point will have relevance for readers of our study over the next years, as our study for the first time directly addresses the numerous requests from the oncological community to gain better biological rationale of radiomics in general. Indeed, we do not propose to directly replace biospecimen-derived diagnostics with radiomic approaches, but rather augment and complement them in cases where biological measurements are not feasible, such as in situations where frequent treatment monitoring is essential. Furthermore, we aim at contributing to better understanding what the biological basis for the various prognostic associations that have been described for radiomic features over past couple of years. To address this challenging task, we implemented a broad genome-wide analysis with a stringent and thorough statistical design, including external validation of putative associations. We revised our Discussion to reflect the translational impact of our study more clearly.